# Commonsense Reasoning and Knowledge Acquisition to Guide Deep Learning on Robots

**Tiago Mota**[1]**, Mohan Sridharan**[2]
[1] University of Auckland, NZ
[2] University of Birmingham, UK

## Abstract

Approaches based on deep network models are increasingly being used for pattern recognition and decision-making tasks in robotics and AI. These approaches are characterized by a large labeled dataset, high computational complexity, and difficultly in understanding the internal representations and reasoning mechanisms. As a step towards addressing these limitations, our architecture uses non-monotonic logical reasoning with incomplete commonsense domain knowledge, and inductive learning of previously unknown state constraints, to guide the construction of deep networks based on a small number of training examples. As an illustrative example, we consider a robot reasoning about the stability and partial occlusion of object configurations in simulated images of an indoor domain. Experimental results indicate that in comparison with an architecture based just on deep networks, our architecture improves reliability, and reduces the sample complexity and time complexity of training the deep networks.

## 1 Introduction

Consider an assistive robot[1] tasked with clearing away toys that children have arranged in different configurations in different rooms. This task poses a challenging scene understanding problem. It is difficult to provide many labeled examples of different arrangements of objects. In addition, the robot has to reason with different descriptions of uncertainty and incomplete domain knowledge. Information about the domain may include qualitative descriptions of commonsense knowledge, e.g., statements such as "structures with a larger object placed on a smaller object are typically unstable", which hold in all but a few exceptional circumstances. At the same time, algorithms for sensing and navigation may represent uncertainty quantitatively, e.g., using probabilities. Furthermore, human participants may not have the time and expertise to interpret sensor data or provide comprehensive feedback, and reasoning with incomplete knowledge may result in incorrect or sub-optimal outcomes.

State of the art methods for scene understanding are based on deep (neural) networks. Although these methods provide high accuracy for pattern recognition and decision making

---

[1]Terms "robot", "learner", and "agent" used interchangeably.

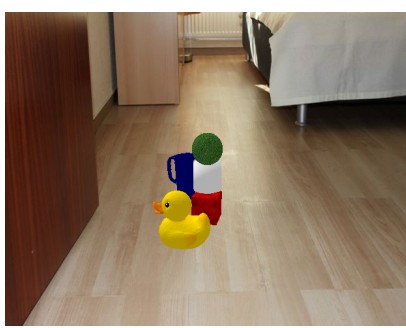

Figure 1: Simulated scene of toys in a room. The robot has to reason about partial occlusion and stability of structures.

tasks in robotics and AI, they require many labeled training samples, are computationally expensive, and provide results that are not easily interpretable. Research in cognitive systems indicates that many of these challenges can be addressed by exploiting domain knowledge and the dependencies between knowledge representation, reasoning and learning. Based on this insight, the architecture described in this paper incorporates non-monotonic logical reasoning with incomplete commonsense domain knowledge, and incremental inductive learning of constraints governing domain states, to guide the learning of deep network architectures. As illustrative examples of scene understanding tasks, we consider an assistive robot estimating the *partial occlusion* of objects and the *stability* of object configurations. To focus on the interplay between representation, reasoning, and learning, we consider simulated images of scenes, e.g., Figure 1, and limit perceptual processing to that of 3D point clouds extracted from the scene. We also assume that the robot knows the grounding (i.e., meaning in the physical world) of words such as "above" and "left_of" that describe basic geometric relations between domain objects. We then describe the following features of our architecture:

- Non-monotonic logical reasoning is used to perform the estimation tasks on each input image based on the commonsense domain knowledge and the geometric relations between objects extracted from the image.

- A small set of labeled examples, i.e., images with occlu-

sion and stability labels for objects and object structures, is used to train decision trees and incrementally learn previously unknown constraints governing domain states.

- Regions of images for which non-monotonic logical reasoning is unable to perform the estimation tasks, are identified automatically and used to train deep network models or processed using the learned models during testing.

Experimental results show a marked improvement in accuracy and computational efficiency in comparison with just using deep networks, while also providing insights about the interplay between reasoning and learning. Section 2 discusses related work, and Section 3 describes the architecture. Experimental results are discussed in Section 4 and the conclusions are in Section 5.

## 2  Related Work

Scene understanding is a key problem in computer vision and robotics; it involves the estimation of relations between scene objects and other prediction problems. Algorithms based on deep networks represent state of the art for many scene understanding, computer vision and control problems. For instance, a Convolutional Neural Network (CNN) has been used to predict the stability of a tower of blocks [16], and to predict the movement of an object sliding down an inclined surface and colliding with another object [30]. However, CNNs and other deep networks require many labeled examples to learn the mapping from inputs to outputs Also, they are computationally expensive, their operation is not easily interpretable, and it is difficult to transfer knowledge learned in one scenario or task to another [31]. In dynamic domains in which it is difficult to obtain many labeled examples, one popular approach is to use physics engines, e.g., for using deep networks to predict the movement of objects when external forces are applied [20].

Prior knowledge has been used to reduce the computational effort and the need for many labeled examples in training deep networks [29]. An RNN augmented by arithmetic and logic operations has been used to answer questions about the scene, but it used textual data instead of visual data [21]. Prior knowledge has also been used to encode state constraints in the CNN loss function, reducing the effort in labeling images, but the constraints have to be encoded manually [28]. The structure of deep networks has also been used to constrain learning, e.g., by using relational frameworks that consider pairs of objects and related questions for visual question answering [24]. This approach, however, only makes limited use of the available knowledge, and does not revise the knowledge over time.

For scene understanding, domain knowledge often includes the grounding (i.e., interpretation in the physical world) of spatial relations such as *in*, *behind*, and *above*. Measures related to the relative position of objects have been used to predict the successful application of actions in a new scenario [7], and researchers have explored reasoning about and learning spatial relations between objects [10, 17]. Deep networks have been used to infer spatial relations between scene objects from images and natural language expressions for manipulation [22], navigation [23] and HRI [25].

There is much work in AI on learning domain knowledge. Early work used a first-order logic representation and incrementally refined the action operators [9]. More recent work used inductive learning to acquire domain knowledge represented as Answer Set Prolog (ASP) programs [14], and integrated non-monotonic logical reasoning and relational reinforcement learning to incrementally learn domain axioms [27]. These learning approaches may be viewed as instances of interactive task learning, a framework for acquiring domain knowledge using labeled examples or reinforcement signals obtained from domain observations, demonstrations, or human instructions [4, 13]. These methods build on early work on joint search through the space of hypotheses and observations [26], but such methods have not been fully explored for scene understanding.

In this paper, we assume that grounding of spatial relations is computed using our prior work [18], and explore the complementary strengths of deep learning, non-monotonic logical reasoning with commonsense knowledge, and incremental learning of the domain's state constraints.

## 3  Proposed Architecture

Figure 2 is an overview of our architecture. Inputs include RGB-D images of scenes with different object configurations, and some commonsense domain knowledge. During training, inputs also include occlusion labels of objects and stability labels of object structures in a small number of images. An existing method is used to ground the spatial relations between objects [18]. An object is said to be occluded if the view of any fraction of its frontal face is hidden by another object, and a structure is unstable if any component object is unstable. Domain knowledge, object attributes, and the spatial relations are encoded in an ASP[2] program. If ASP-based reasoning provides the desired labels, no further analysis of this image is performed. Otherwise, an attention mechanism uses domain knowledge to identify the image's Regions of Interest (ROIs), with one or more objects in each ROI. A CNN is trained to map each ROI to labels. The spatial relations, object attributes, and labels are also used to incrementally learn a decision tree and construct axioms denoting state constraints that are added to the ASP program. During testing, the input image is either processed by ASP-reasoning or (if that fails) the learned CNN. Individual modules are described using the following illustrative domain.

**Example 1** *[Robot Assistant (RA)] A simulated robot has to estimate occlusion of objects and stability of object structures in images with toys in different configurations. Robot also has to rearrange objects to reduce clutter. An object's attributes include* size *(small, medium, large),* surface *(flat, irregular) and* shape *(cube, cylinder, duck). The* relation *between objects can be (above, below, front, behind, right, left, close). The robot can move objects to achieve assigned goals. Domain knowledge includes axioms governing dynamics but some axioms may be unknown, e.g.:*

- *Placing an object on top of an object with an irregular surface causes instability;*

---

[2]We use "ASP" and "CR-Prolog" interchangeably.

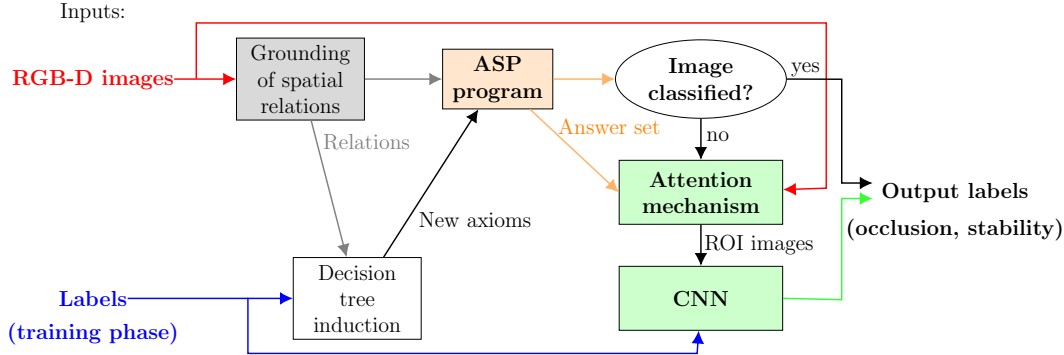

Figure 2: Proposed architecture combines the complementary strengths of non-monotonic logical reasoning, deep learning and decision tree induction to perform the scene understanding tasks reliably and efficiently.

- *Removing all objects in front of an object causes this object to not be occluded.*

### 3.1 Knowledge Representation with ASP

To represent and reason with incomplete domain knowledge, we use ASP, a declarative language that can represent recursive definitions, defaults, causal relations, and language constructs that occur frequently in non-mathematical domains and are difficult to express in classical logic formalisms. ASP supports concepts such as *default negation* (negation by failure) and *epistemic disjunction*, e.g., unlike "¬a", which implies that "*a is believed to be false*", "not a" only implies "*a is not believed to be true*". Each literal can be true, false or unknown and the *robot only believes things that it is forced to believe*. ASP supports non-monotonic logical reasoning, i.e., adding a statement can reduce the set of inferred consequences, aiding in the recovery from errors due to the incomplete knowledge [8]. ASP and other similar paradigms are often criticized for requiring considerable prior knowledge, and for being unwieldy in complex domains. However, modern ASP solvers support efficient reasoning in large, incomplete knowledge bases, and are used by an international research community for many applications [5, 6].

A domain description in ASP comprises a *system description* $\mathcal{D}$ and a *history* $\mathcal{H}$. $\mathcal{D}$ comprises a *sorted signature* $\Sigma$ and axioms. $\Sigma$ comprises *sorts* arranged hierarchically; *statics*, i.e., domain attributes that do not change over time; *fluents*, i.e., domain attributes whose values can be changed; and *actions*. In the RA domain, sorts include $object$, $robot$, $size$, $relation$, $surface$ and $step$ (for temporal reasoning). Statics include object attributes such as $obj\_size(object, size)$ and $obj\_surface(obj, surface)$. Spatial relations $obj\_rel(relation, object, object)$ between objects are fluents described in terms of their arguments' sorts, e.g., $obj\_rel(above, A, B)$ implies object $A$ is *above* object $B$. The last argument in these relations is the reference object. Besides spatial relations, fluents describe other aspects of the domain, e.g., $in\_hand(robot, object)$ and $stable(object)$, while $pickup(robot, object)$ and $putdown(robot, object)$ are actions. Also, predicate $holds(fluent, step)$ implies that a particular fluent holds true at a particular timestep.

The axioms of $\mathcal{D}$ govern domain dynamics and include:

$$holds(in\_hand(robot, object), I + 1) \leftarrow \qquad (1a)$$
$$occurs(pickup(robot, object), I)$$

$$holds(obj\_rel(above, A, B), I) \leftarrow \qquad (1b)$$
$$holds(obj\_rel(below, B, A), I)$$

$$\neg occurs(pickup(robot, object), I) \leftarrow \qquad (1c)$$
$$holds(in\_hand(robot, object), I)$$

where Statement 1(a) describes a causal law, (b) describes a constraint, and (c) describes an executability condition. The spatial relations extracted from RGB-D images are converted to facts used in ASP program. The program also includes axioms that encode default knowledge, e.g., "larger objects on smaller objects are typically unstable".

$$\neg holds(stable(A), I) \leftarrow holds(obj\_rel(above, A, B), I),$$
$$size(A, large), \; size(B, small),$$
$$not \; holds(stable(A), I) \qquad (2)$$

Finally $\mathcal{H}$ includes records of observations received and actions executed by the robot. To reason with the existing knowledge, we construct CR-Prolog program $\Pi(\mathcal{D}, \mathcal{H})$—please see our code repository [19]. Planning, diagnostics and inference tasks can then be reduced to computing *answer sets* of $\Pi$, which represent beliefs of the robot associated with $\Pi$ [8]. We use SPARC [2] to compute answer set(s).

### 3.2 Decision Tree Induction

The spatial relations identified between pairs of objects and the attributes of objects are used to build decision trees for classification. In the RA domain, separate decision trees are built for estimating stability and occlusion, with the labels assigned to the leaf nodes being stable/unstable or occluded/not occluded respectively. One half of the available examples are used for training, while the other half is used for validating the axioms extracted from learned trees. We used an existing tree induction algorithm that computes the entropy (and thus information gain) of a split in the tree based on each attribute. Illustrative examples of decision trees are shown in Figures 3 and 4.

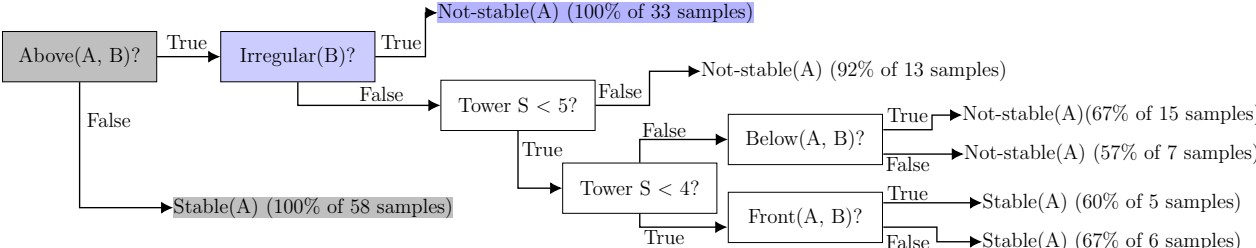

Figure 3: Example of a decision tree constructed for stability estimation using some labeled examples. Highlighted branches are used to construct previously unknown axioms.

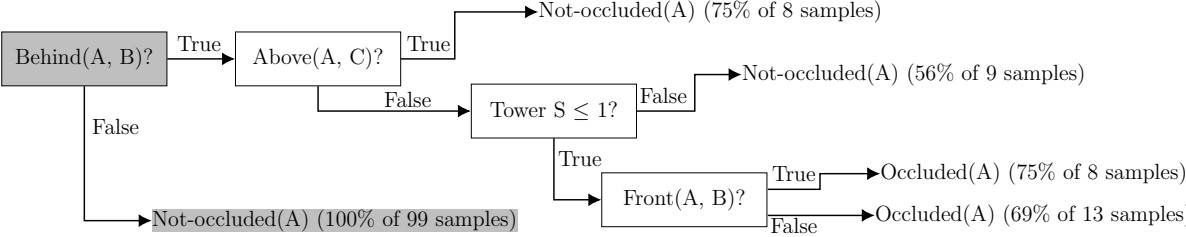

Figure 4: Example of a decision tree constructed for occlusion estimation using some labeled examples. Highlighted branch is used to construct previously unknown axiom.

Any branch of a decision tree in which the leaf represents a precision higher than $95\%$, i.e., most examples correspond to a particular class, is used to construct axioms that are analyzed using the validation set. This process: (i) removes axioms without a minimum level of support; and (ii) merges the discovered axioms to retain them in their most general form. Since the number of labeled examples is small, we reduce the effect of noise by repeating the learning and validation process a number of times (e.g., 100). Axioms voted more than a minimum number of times (e.g., $50\%$) are encoded in the ASP program and used for reasoning.

Consider the branches highlighted in gray in Figures 3 and 4, which can be translated into the following axioms:

$$stable(A) \leftarrow \neg obj\_rel(above, A, B) \tag{3a}$$
$$\neg occluded(A) \leftarrow \neg obj\_rel(behind, A, B) \tag{3b}$$

where Statement 3(a) implies that any object that is not above another object is stable, and Statement 3(b) says that an object is not occluded if it is not located behind another object. More elaborate axioms are created when other object attributes (e.g., size, surface) are considered, e.g., the branch highlighted in gray and blue in Figure 3 is:

$$\neg stable(A) \leftarrow obj\_rel(above, A, B), \tag{4}$$
$$obj\_surface(B, irregular)$$

which says that an object is unstable if it is located above an object with an irregular surface. The architecture is also able to discover axioms corresponding to default knowledge by lowering the threshold for selecting a branch of a tree (e.g., to $70\%$), but this also introduces noisy estimates of axioms.

### 3.3 Attention Mechanism

The attention mechanism module is only invoked to process an input image if ASP-based reasoning is unable to assign labels to the objects in the image. This module then identifies the regions of interest (ROIs) in the image that need to be analyzed further. More specifically, the module first identifies axioms in the ASP program whose head corresponds to a relation or fluent of interest. For instance, if the robot's task is to estimate the stability of object configurations, the attention mechanism will identify Statement 3(a) and Statement 4, which define conditions under which an object is considered to be stable or unstable (respectively). In a similar manner, Statement 3(b) will be considered when the task is to examine the occlusion of objects. For each axiom considered to be of interest, the body of the axiom provides the relations to be used to identify ROIs in the corresponding image; other image regions are unlikely to provide useful information and are thus not analyzed further. For instance, while estimating stability in Figure 1, we should consider the stack comprising the red cube, white cylinder and the green ball, since they satisfy a relevant relation ($above$)— the other two objects (duck and pitcher) can be disregarded. Any image may contain multiple such ROIs, and each ROI may have multiple objects.

### 3.4 Convolutional Neural Networks

The ROIs identified by the attention mechanism serve as input to a deep network—we use two variants of a CNN. Recall that pixels of any such ROI contains information directly relevant to the task at hand. The training dataset for the CNN also includes labels to be assigned to objects in the ROIs. The CNN learns the mapping between the image pixels and

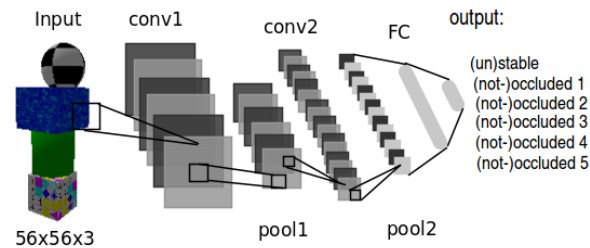

Figure 5: Lenet architecture.

labels, and assigns labels to ROIs in previously unseen test images that ASP-based reasoning is unable to process.

CNNs can vary in terms of the number of layers, activation functions etc, but the building blocks are convolutional and pooling layers used in the initial or intermediate stages, and fully-connected layers that are typically one of the final layers. In a convolutional layer, a filter (or kernel) is convolved with the original input or the output of the previous layer. Common pooling strategies are max-pooling and average-pooling, which are used to reduce the dimensions of the input data and control overfitting. One or more convolutional layers are usually followed by a pooling layer. The fully-connected layers are equivalent to feed-forward neural networks; they often provide the target output(s). In the context of images, convolutional layers extract useful attributes to model the mapping from inputs to outputs. For instance, the initial layers may extract lines and arches, whereas the subsequent layers may represent complex shapes. In the particular context of estimating the stability of object configurations, the CNN's layers may represent whether a tower of blocks is aligned, or if a tower has a small base.

In this paper, we adapted two CNN architectures: the simple Lenet [15], initially proposed for recognizing handwritten digits; and the widely used Alexnet [12], which provided best results on the Imagenet (benchmark) challenge in 2012. The Lenet has two convolutional layers, each one followed by a max-pooling layer and an activation layer. Two fully connected layers are placed at the end. Unlike the $28 \times 28$ gray-scale input images and the ten-class softmax output layer used in the original implementation for classifying digits, we consider $56 \times 56$ RGB images as input and an output vector representing the occlusion and stability of each object in the image. Figure 5 is a pictorial representation of this network—as described later, we consider ROIs with up to five objects in the experimental studies. The Alexnet architecture, on the other hand, contains five convolutional layers, each followed by max-pooling and activation layers, along with three fully connected layers at the end. In our experiments, $227 \times 227$ RGB images were used as input and the output classes determined the target variables estimating occlusion and stability. We have five outputs estimating occlusion to consider ROIs with up to five objects, and one output for stability of the scene. Due to the multi-class labeling problem, the sigmoid activation function was used in both networks. We used the Adam optimizer [11] in TensorFlow [1] with a learning rate of 0.0001 for the Alexnet network and 0.0002 for the Lenet network and the weights were initialized randomly. The number of training iterations varied depending on the network and the number of training examples. For example, Lenet using 100 and 5,000 image samples was trained for 10,000 and 40,000 iterations, respectively, whereas the Alexnet with 100 and 5,000 samples was trained for 8,000 and 20,000 iterations, respectively. The learning rate and number of iterations were chosen experimentally using validation sets. The number of epochs was chosen as the stopping criteria, instead of the training error, in order to allow the comparison between networks learned with and without the attention mechanism. The code for training the deep networks is in our repository [19].

## 4    Experimental Setup and Results

In this section, we describe the experimental setup and the results of experimental evaluation of our architecture.

### 4.1    Experimental Setup

For experimental evaluation, we simulated a domain in which many labeled examples are not available. We generated 6000 labeled images using a real-time physics engine and constructed a dataset for estimating occlusion and stability of objects. Each image had ROIs with up to five objects of different colors, textures and shapes. The objects included cylinders, spheres, cubes, a duck, and five household objects from the Yale-CMU-Berkeley dataset (apple, pitcher, mustard bottle, mug, and cracker box) [3]. These objects are arranged in three configurations:

- **Towers**: $2 - 5$ objects stacked on each other;
- **Spread**: five objects on the flat surface (ground); and
- **Intersection**: $2-4$ objects stacked on each other, with the rest $(1 - 3)$ on the flat surface.

The vertical alignment of stacked objects is randomized to create stable or unstable arrangements. The horizontal distance between objects is also randomized to create scenes with complex, partial or no occlusion. Lighting, orientation, camera distance, orientation, and background, were also randomized. The corresponding ASP program was initially missing three state constraints (each) related to stability estimation and occlusion estimation.

A second dataset was then derived from the dataset described above to simulate the attention mechanism. Recall that this module extracts suitable ROIs from images in the original dataset for which ASP-based reasoning is unable to assign labels. Pixels in these images that are outside the ROI are cropped off. CNNs trained using these two datasets were compared as a function of the amount of training data and the complexity of the networks. Occlusion was estimated for each object (five outputs) and stability was estimated for the structure (1 output). The experiments were designed to test the following hypotheses:

- **H1:** Reasoning with commonsense domain knowledge and the associated attention mechanism improves the accuracy of deep networks.
- **H2:** Reasoning with commonsense domain knowledge and the attention mechanism reduces sample complexity and time complexity of training deep networks.

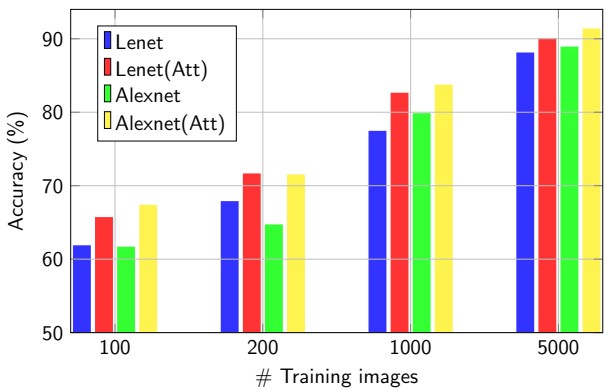

Figure 6: Accuracy of Lenet and Alexnet CNNs improves significantly with the commonsense reasoning module and the attention mechanism.

- **H3:** The architecture is able to incrementally learn previously unknown axioms, and use these axioms to improve the accuracy of decision making.

The performance measures were the accuracy of the labels assigned to objects and structures in images, and the number of samples and time taken to train the networks. Below, *all claims are statistically significant at the* $95\%$ *significance level*. As the baseline for comparison, we trained and tested the Lenet and Alexnet architectures without the commonsense reasoning and attention mechanism modules, i.e., directly on the RGB-D input images.

### 4.2 Experimental Results

The first set of experiments was designed as follows, with results summarized in Figure 6:

1. Training datasets of different sizes (100, 200, 1000, and 5000 images) were used to train the Lenet and Alexnet networks. The remaining images were used for testing. Attention mechanism and commonsense reasoning were not used, with results summarized as "Lenet" and "Alexnet" in Figure 6.

2. The datasets after applying the attention mechanism were derived from the datasets in step-1, and used to train and test the Lenet and Alexnet networks, with the results plotted as "Lenet(Att)" and "Alexnet(Att)" in Figure 6.

Figure 6 indicates that our architecture improves the accuracy of the Lenet and Alexnet networks for the joint estimation of stability and occlusion in scenes. We notice that training and testing the deep networks with only those images that ASP-based reasoning cannot label, enables attention to be focused on the relevant image regions, resulting in better performance. The benefits are more pronounced when the training dataset is smaller, but there is significant improvement in performance at all training dataset sizes. These results support hypothesis **H1**.

Figure 7 shows two examples of the improvement provided by the attention mechanism. In Figure 7a, both *Lenet* and the *Lenet(Att)* networks were able to recognize the occlusion of the *red cube* caused by the *green mug*, but only

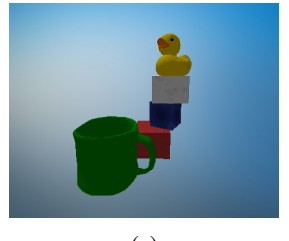
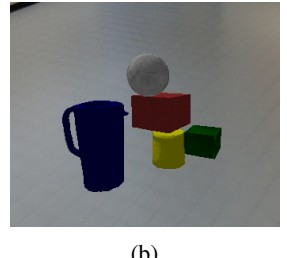

(a)       (b)

Figure 7: Examples of test images for the Lenet CNN: (a) Lenet and Lenet(Att) detected the occlusion of the red cube by the green mug, but only Lenet(Att) correctly predicted the tower's instability; and (b) Lenet and Lenet(Att) predicted the instability of the tower, but only the Lenet(Att) detected the occlusion of the green cube by the yellow cylinder.

Lenet(Att), which uses the attention mechanism in conjunction with commonsense reasoning, was able to estimate the instability of the tower. In Figure 7b, both networks correctly predicted the instability of the tower, but only *Lenet(Att)* identified the occlusion of the *green cube* by the *yellow can*. The classification errors are most probably because a similar example had not been observed during training—the inability to identify the true cause of the error is a known limitation of deep architectures. The attention mechanism focuses the attention of the network on the relevant image regions, resulting in better classification accuracy. For this example, the CNNs were trained with 1000 images, and the two test scenes were not seen during training.

The second set of experiments was designed as follows, with results summarized in Figure 8:

1. The Lenet network was trained with training datasets containing between $100 - 1000$ images, in step-sizes of 100. Separate set of scenes was created for testing. The baseline CNN used the training datasets without the commonsense reasoning module or attention mechanism.

2. The dataset after applying the attention mechanism was derived from these training datasets, and used to train and test a CNN, with the results plotted as "Lenet(Att)" in Figure 8.

In these experiments, we only used *Lenet* and not *Alexnet* because the former was observed to provide performance comparable to the latter but with much less computational effort.

Figure 8 indicates that the attention mechanism supported by commonsense reasoning achieves a desired level of accuracy with much fewer training examples. For instance, the orange dashed line in Figure 8 indicates that the baseline Lenet needs $\approx 1000$ images to reach an accuracy of 77%, whereas our architecture reduces this number to $\approx 600$. In other words, the deep networks can be trained with fewer examples because the commonsense knowledge is exploited, reducing both the computation and storage requirements. These results support hypothesis **H2**.

Finally, the third set of experiments was designed as follows, with results summarized in Table 1:

1. Ten sets of 50 labeled images were created by random

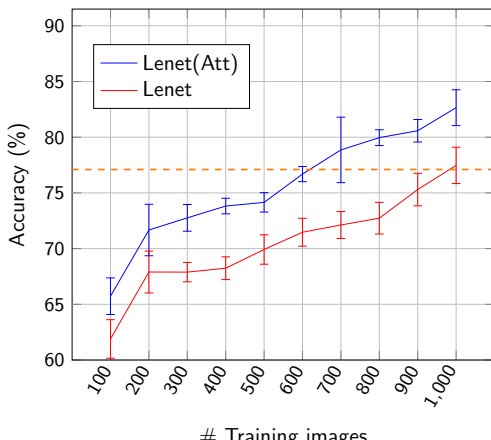

Figure 8: Accuracy of Lenet with and without the attention mechanism and commonsense reasoning module. Any desired accuracy is achieved with a much smaller training set.

| Axioms | Precision | Recall |
|---|---|---|
| Unknown (normal) | 98% | 100% |
| Unknown (default) | 78% | 62% |

Table 1: Precision and recall for unknown axioms (normal, default) using decision tree induction.

selection. The axiom learning algorithm was trained with each set three times, using thresholds of 95% and 70% as described in Section 3.2.

2. The precision and recall for the unknown axioms (with threshold of 95%), e.g., Statements 3(a), 3(b), and 4, are summarized as "unknown (normal)" in Table 1.

3. The precision and recall for the unknown default statements (with threshold of 70%), e.g., Statement 2, are summarized as "unknown (default)" in Table 1;

Table 1 demonstrates the ability to learn previously unknown axioms. Errors are predominantly variants of the target axioms that are not in the most generic form, i.e., they have some irrelevant literals. The lower precision and recall with defaults is expected because it is challenging to distinguish between defaults and their exceptions. Although we do not describe it here, reasoning with commonsense knowledge and decision trees also provides explanations for the architecture's performance.

Finally, we ran experiments with the objective of computing minimal plans to pickup and clear particular objects. The number of plans computed when the learned axioms are included in the ASP program is much smaller than when the axioms are not included—the learned axioms are constraints that eliminate irrelevant paths in the transition diagram. For instance, the goal in one experiment was to clear the large red box partially hidden behind the white box and the duck in Figure 9. With all the axioms, eight plans are found (all of which were correct); with some axioms missing, as many as 90 plans are found, many of which were incorrect. These

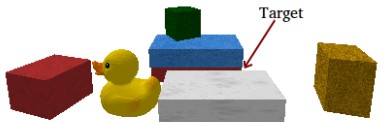

Figure 9: Illustrative image used for planning experiments with and without the learned axioms.

results support hypothesis **H3**.

## 5   Conclusion

Deep network architectures and algorithms are providing state of the art performance for many pattern recognition tasks in robotics and AI. However, they require large training datasets and considerable computational resources, and make it difficult to understand their operation. The architecture described in this paper draws inspiration from research in cognitive systems to address these limitations. It combines the principles of reasoning with incomplete commonsense domain knowledge, and decision tree induction, with deep learning. In the context of estimating occlusion of objects and the stability of object configurations in simulated images, we observe that the proposed architecture improves the accuracy and computational efficiency of the deep network architectures. Future work will further examine the performance of this architecture in more complex domains, and explore the explainability of the observed performance.

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
