# OpenReview forum: "Commonsense Reasoning and Knowledge Acquisition to Guide Deep Learning on Robots"
_icaps-conference.org/ICAPS/2019/Workshop/KEPS — KEPS 2019_

### Official Review · AnonReviewer3 · 2019-05-08
**Interesting approach for knowledge acquisition in the context of object occlusion and stability estimation for robotic application**

**Rating:** 4
**Confidence:** 2

**Review:**

The paper is well written and addresses an interesting problem in robotics and knowledge engineering. The results look sound and show the improved accuracy and computational efficiency of the proposed architecture and the deep
network system. The only comment I would say is to define all the acronyms used in the paper (e.g. RNN) for completeness.

---

### Official Review · AnonReviewer1 · 2019-05-15
**Not well suited for KEPS**

**Rating:** 2
**Confidence:** 3

**Review:**

The paper presents a classical object recognition problem and proposes to use a solution leveraging the integration of commonsense knowledge/reasoning and learning. Attentional mechanisms are proposed to improve the accuracy of the proposed approach. An experimental evaluation (with 3d simulations) is presented and discussed.

In my opinion, some major issues affect the paper:
- The paper does not seem to consider significant Knowledge Engineering issues and, most importantly, there is no clear connection to Planning and Scheduling.
- A wide amount of work has been done on object recognition/classification. The related works section in the paper is strongly focused on (rather recent) learning approaches and does not seem to provide a complete analysis of state of the art. For instance, work on cognitive robotics by, e.g., Michael Beetz or Joachim Hertzberg (just to mention few relevant researchers) are not considered. A wider analysis of related works is required also to better situate the claimed contribution.
- The paper does not clearly describes how commonsense reasoning and knowledge-based mechanisms are actually exploited to improve learning efficacy.
- References to a service robot are mentioned but I can't see a concrete relationships with the proposed approach. The involvement of a robot does not seem actually crucial to me for the proposed approach. And the paper does not bring any clear evidence of how such embodiment can be exploited.

In general, the paper does not seems to provide a crisp and significant contribution for KEPS. Related works should be better investigated and discussed. The exploitation of commonsense reasoning and knowledge-based mechanisms should be better described. And, last but not least, planning and scheduling issues seem to be not considered. Thus, the paper does not sound to me very well suited to KEPS audience.